# C-terminal tagging, transmembrane domain hydrophobicity, and an ER retention motif influence the secretory trafficking of the inner nuclear membrane protein emerin

Jessica Mella[1], Regan F Volk[1], Balyn W Zaro[1,2], Abigail Buchwalter[1,3]*

[1]Cardiovascular Research Institute, University of California, San Francisco, San Francisco, United States; [2]Department of Pharmaceutical Chemistry, University of California, San Francisco, San Francisco, United States; [3]Department of Physiology, University of California, San Francisco, San Francisco, United States

## eLife Assessment

This study presents a **valuable** finding on the delivery of a nuclear envelop protein to lysosomes and the impact of C-terminal tagging on its traffic. The authors provide **solid** evidence for the potential artifacts introduced by large terminal tags, particularly in the context of membrane protein localization and stability.

**Abstract** The inner nuclear membrane (INM), a subdomain of the endoplasmic reticulum (ER), sequesters hundreds of transmembrane proteins within the nucleus. We previously found that one INM protein, emerin, can evade the INM by secretory transport to the lysosome, where it is degraded (Buchwalter et al., 2019). In this work, we used targeted mutagenesis to identify intrinsic sequences that promote or inhibit emerin's secretory trafficking. By manipulating these sequences across several tag and expression level combinations, we now find that emerin's localization is sensitive to C-terminal GFP tagging. While emerin's long, hydrophobic C-terminal transmembrane domain facilitates trafficking to the lysosome, extending its lumenal terminus with a GFP tag biases the protein toward this pathway. In contrast, we identify a conserved ER retention sequence that stabilizes N- and C-terminally tagged emerin by limiting its lysosomal flux. These findings underscore long-standing concerns about tagging artifacts and reveal novel determinants of tail-anchored INM protein targeting.

*For correspondence:
abigail.buchwalter@ucsf.edu

Competing interest: The authors declare that no competing interests exist.

## Introduction

Tail-anchored proteins of the inner nuclear membrane (INM) are a unique class of transmembrane proteins with poorly understood biosynthesis, targeting, and degradation. The INM is a specialized extension of the endoplasmic reticulum (ER) membrane, the major site of membrane protein synthesis. Shortly after translation, tail-anchored proteins are embedded by their C-terminal transmembrane domains (TMDs) into the ER (*Rapaport and Herrmann, 2023*). Unlike cytoplasmic tail-anchored proteins, which rapidly traffic out of the ER toward their target membranes, INM proteins must remain in the ER long enough to diffuse through the nuclear pore complex and be retained by

binding to nuclear structures (*Boni et al., 2015*; *Ungricht et al., 2015*). This process repeats each time the nuclear envelope reforms at the end of mitosis (*Ellenberg et al., 1997*). INM proteins thus spend considerable time in the ER, but how they interact with the ER's secretory or proteostatic functions remains unclear.

Our previous work identified emerin (EMD) as a short-lived INM protein in C2C12 mouse myoblasts (*Buchwalter et al., 2019*). While investigating how emerin is degraded, we found that C-terminally GFP-tagged emerin travels by vesicular transport to the Golgi, plasma membrane (PM), and then lysosome during ER stress (*Buchwalter et al., 2019*) – an itinerary that closely tracks with the path taken by cargoes of the rapid ER stress-induced export pathway (*Satpute-Krishnan et al., 2014*). This unexpected secretory trafficking was prevented by the removal of EMD-GFP's N-terminal LEM domain, suggesting that this domain played a role in stress-induced lysosomal degradation. We hypothesized that the INM might dynamically respond to ER stress by shunting emerin through the secretory pathway to the lysosome. However, we now find that N-terminally GFP-tagged emerin traffics at a lower rate than C-terminally tagged emerin and find no conclusive evidence that untagged emerin traffics to the lysosome. We also show that the secretory trafficking of tagged emerin depends on its unusually hydrophobic TMD. In isolation, the TMDs of both emerin and the related tail-anchored protein LAP2β readily traffic to the PM, but trafficking of the full-length tagged proteins is limited by a conserved RXR-type ER retention motif. Our experiments indicate that emerin's trafficking is artifactually expedited by C-terminal GFP tagging. However, these studies identify sequence elements that target emerin to the contiguous nuclear envelope/endoplasmic reticulum (NE/ER) network and highlight the interdependence between protein biogenesis, topology, and localization.

## Results

After our initial discovery, we aimed to understand the mechanism of emerin's secretory trafficking and its conservation across species, cell types, and expression levels. As our earlier experiments used C2C12 mouse myoblasts, we first tested whether EMD-GFP trafficking is conserved in human cells. We stably expressed tet-inducible mouse and human GFP fusions in human osteosarcoma (U2OS) cells. When treated with the lysosome blocking agent bafilomycin A1 (BafA1), we found that U2OS cells, unlike C2C12 myoblasts (*Buchwalter et al., 2019*), do not rely on ER stress to initiate secretory trafficking of EMD-GFP. Instead, BafA1 alone caused EMD-GFP to accumulate in lysosomal puncta. We used this high secretory flux to differentiate between emerin constructs that could and could not traffic (*Figure 1A*). Consistent with our C2C12 findings, we verified that WT EMD-GFP could traffic to the lysosome of U2OS cells, while a construct lacking the N-terminal LEM domain (ΔLEM, *Figure 1B*) could not. Our constructs were tagged with a C-terminal/lumenal GFP, which allowed us to quantify the proportion of EMD-GFP in the secretory pathway using cell surface antibody labeling and flow cytometry (*Figure 1C and D*). We calculated the trafficking level of each construct by dividing the surface anti-GFP antibody signal by the total EMD-GFP fluorescence. In line with our microscopy experiments, the anti-GFP antibody did not detect any ΔLEM construct at the PM of U2OS cells, while the WT EMD-GFP PM signal was always above background (*Figure 1E*).

### Emerin's hydrophobic TMD is necessary and sufficient for trafficking of EMD-GFP

We further leveraged this U2OS system to examine how emerin's protein sequence facilitates trafficking. Emerin contains three major domains: an N-terminal LEM domain that interacts with the chromatin cross-linker BANF1/BAF, an intrinsically disordered region that binds the nuclear lamina, and a C-terminal TMD (*Lee et al., 2001*; *Figure 1B*). We tested each domain's ability to traffic through the secretory pathway with a series of domain truncation experiments. Having established that removing the LEM domain abolished emerin's ability to traffic, we next deleted the disordered region between the LEM domain and the TMD. This construct (termed LEM-TMD) accumulated at the PM 10-fold more than the WT protein, indicating that the TMD, LEM domain, or both together are sufficient for trafficking (*Figure 1F*). To distinguish between these possibilities, we tested whether emerin's TMD alone can travel to the PM. We removed all but the TMD alpha helix and its surrounding 26 residues (mEMD 212–259), which we found to be required for membrane insertion of the truncated construct (not shown), and appended this TMD-only construct to either the C- or N-terminus of eGFP. Both

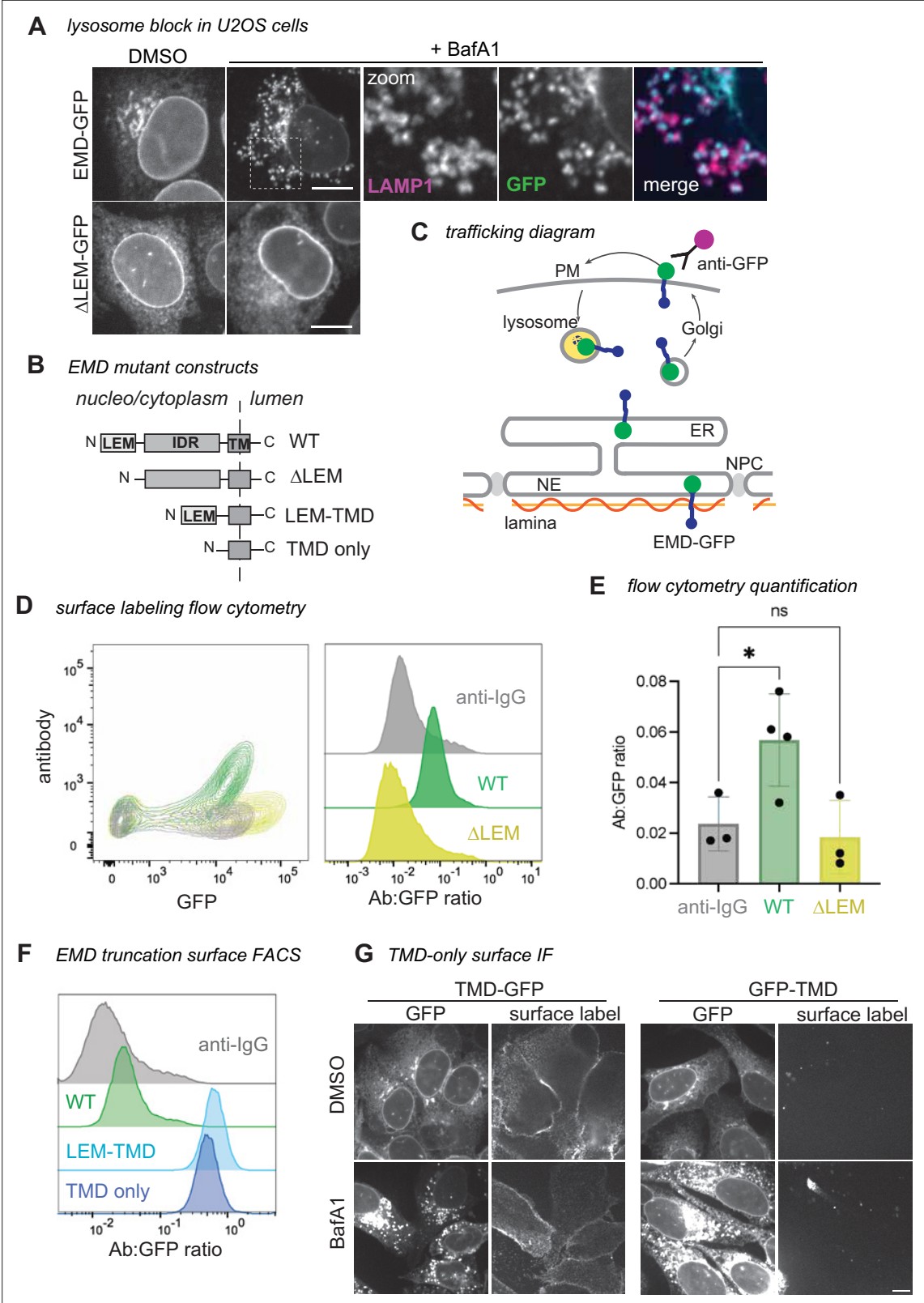

**Figure 1.** EMD ΔLEM does not traffic in U2OS cells, while the transmembrane domain (TMD) alone does. (**A**) U2OS cells expressing mouse EMD-GFP and ΔLEM-GFP were induced with 1 µg/mL doxycycline for 24 hr, treated with 100 nM lysosome blocking agent bafilomycin A1 (BafA1) overnight, and fixed and stained for lysosome marker Lamp1. Scale bar, 10 µm. (**B**) Summary of EMD domains and truncation mutants. (**C**) EMD enriches at the inner nuclear membrane (INM) by binding to the nuclear lamina or exits the endoplasmic reticulum (ER) into the secretory pathway. Before reaching

*Figure 1 continued on next page*

*Figure 1 continued*

the lysosome, EMD transiently accesses the cell surface where lumenal GFP is exposed to anti-GFP antibody. (**D**) Example FACS plot of mouse WT vs ΔLEM surface labeling; fluorescent anti-IgG secondary antibody included as background control. Antibody signal is divided by total GFP+ signal to yield histograms on the right. (**E**) Quantification of the antibody:GFP ratio from 4 independent experiments, with error bars representing SD. * indicates adjusted p-value=0.0451 by one-way ANOVA with Šídák's multiple comparisons test. (**F**) U2OS cells expressing mouse TMD-GFP (lumenal tag) or GFP-TMD (cytosolic tag) were treated overnight with BafA1, then fixed and stained with an anti-GFP antibody. (**G**) Example surface labeling FACS histogram of U2OS cells expressing WT, LEM-TMD-GFP, or TMD-GFP.

TMD-GFP and GFP-TMD localized prominently to the ER and PM, and both were evident in the lysosome after BafA1 treatment (*Figure 1F and G*). These experiments reveal that emerin's TMD alone can traffic to the lysosome, regardless of GFP tag orientation.

We wondered what properties of emerin's TMD mediate its secretion. Tail-anchored proteins can sort via their TMDs along the secretory pathway (*Ronchi et al., 2008*; *Borgese, 2016*; *Bulbarelli et al., 2002*); long and hydrophobic sequences partition to the thicker, more hydrophobic membranes of the late secretory organelles (*Sharpe et al., 2010*; *Figure 2—figure supplement 1B and C*), while tail-anchored proteins with shorter, less hydrophobic TMDs preferentially remain in the ER. We noticed that emerin's TMD is particularly hydrophobic among other NE/ER TMDs (*Figure 2—figure supplement 1A and B*), which led us to predict that emerin's TMD is required for its exit from the ER. To test this, we replaced emerin's TMD with that of cytochrome B5, a model ER protein with a modestly hydrophobic TMD (*Pedrazzini et al., 2000*). This substitution ablated trafficking through the secretory pathway (*Figure 2A and B*), indicating that emerin's TMD is indeed necessary for trafficking. We tested whether this depended on the hydrophobic content of the alpha helix by sequentially mutating its aromatic residues to alanine. We divided the TMD into N- and C-terminal halves, mutating three aromatic amino acids in each half (*Figure 2C*). The N-terminal TMD mutant was slightly less hydrophobic and had a greater effect on secretory trafficking than the C-terminal mutant (*Figure 2D and E*). Combining the N- and C-terminal mutations additively reduced the surface ratio of EMD-GFP. While these mutations decreased the overall protein expression (*Figure 2—figure supplement 1E*), they did not affect the protein's topology (*Figure 2—figure supplement 1F*), confirming that the observed loss of surface exposure was due to diminished secretory trafficking and not to defects in membrane insertion. These experiments indicate that manipulating the hydrophobicity and/or aromaticity of the TMD influences EMD-GFP's transit to the cell surface.

## Emerin contains a conserved ER retention signal

The autonomous trafficking of emerin's TMD challenged our initial hypothesis that emerin's localization relies on the LEM domain. Because EMD ΔLEM-GFP cannot traffic despite containing the TMD, we wondered whether a sequence within the ΔLEM construct inhibits trafficking. We noted that removing the LEM domain exposed a highly charged and conserved motTif (QRRR) on the N-terminus of the construct (*Figure 2—figure supplement 1D*). Poly-arginine (RXR) motifs are known negative regulators of ER-to-Golgi transit (*Michelsen et al., 2005*) we therefore tested whether this unprotected RXR could explain the ER-restricted localization of ΔLEM-GFP. Indeed, further truncating the ΔLEM construct by deleting the QRRR (ΔLEMΔQRRR) freed the protein from the ER (*Figure 2F–H*).

To test whether the RXR motif limits the trafficking of full-length EMD-GFP, we mutated the QRRR to AAAA (designated RA) and detected more flux to the PM compared to WT EMD-GFP (*Figure 2F and G*). Both ΔLEMΔQRRR-GFP and EMD RA-GFP were less abundant at steady state and were more rapidly degraded after removal of doxycycline from the medium (*Figure 2I and J*), suggesting increased flux to the lysosome. These data indicate that the RXR motif constitutes an ER retention sequence in EMD-GFP that is strengthened by the removal of the LEM domain. We infer that the presence of the LEM domain modulates the accessibility of the RRR motif, while the LEM domain itself is in fact dispensable for trafficking, its removal constitutively exposes the RRR motif, enhancing emerin's ER localization.

We next investigated the interplay between emerin's pro-trafficking TMD and its anti-trafficking RXR motif by cloning the RA mutation into our TMD mutant constructs (see *Figure 2C–E*). The RA mutation generally increased the surface expression of the TMD mutants except for the construct with the lowest hydrophobicity score (TMD^m), which was not significantly altered by the RA mutation (*Figure 3A and B*). We infer from this that the TMD acts upstream of the RXR motif in emerin's

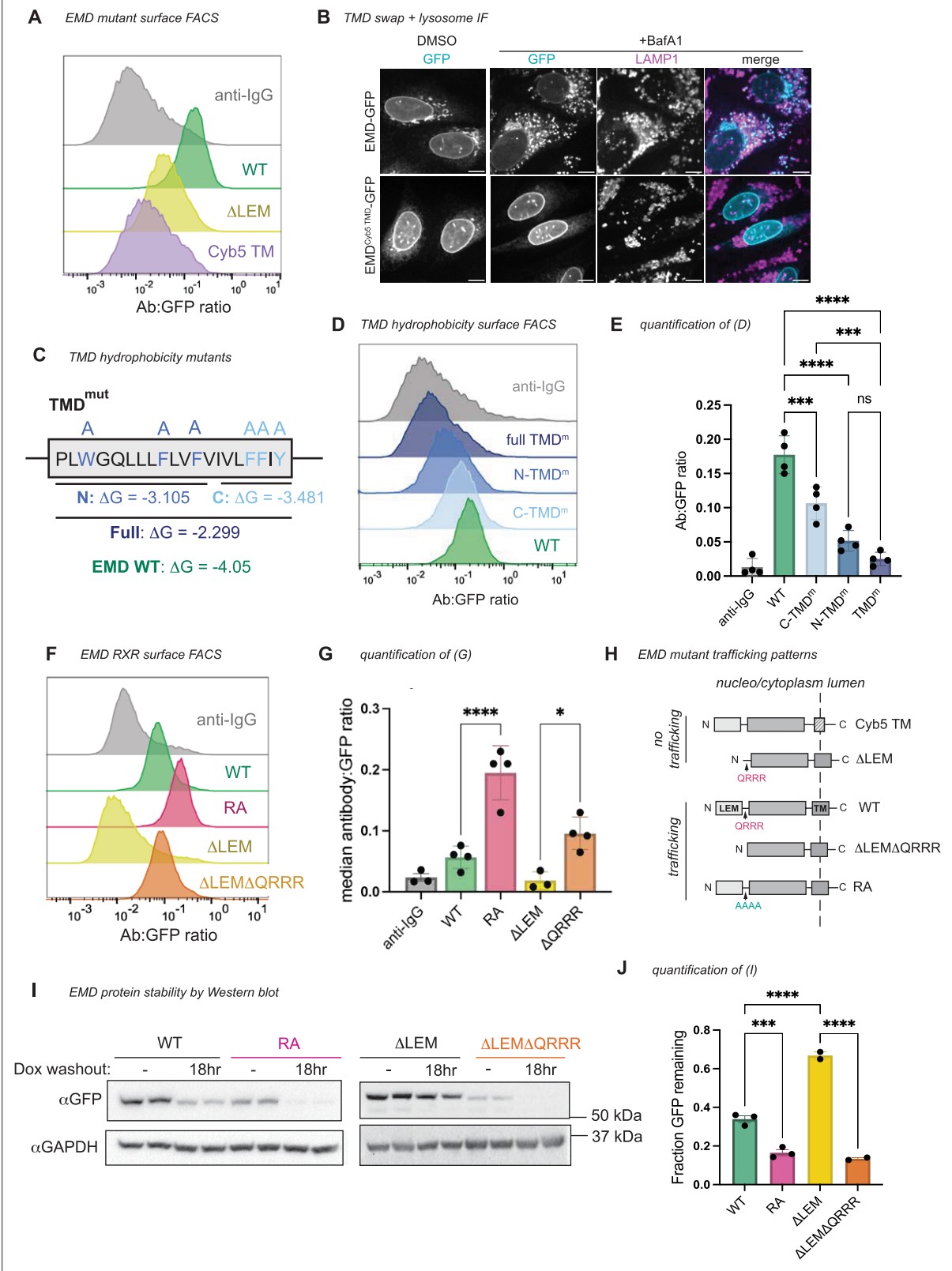

**Figure 2.** Emerin's trafficking depends on its transmembrane domain (TMD), not on the LEM domain. (**A**) Surface anti-GFP antibody:GFP histogram of human WT EMD, ΔLEM, and EMD with mouse cytochrome B5 TMD (EMD$^{Cyb5\,TMD}$) chimera. (**B**) U2OS cells expressing mouse EMD-GFP and EMD$^{Cyb5\,TMD}$-GFP were induced with 1 μg/mL doxycycline for 24 hr, treated with 100 nM lysosome blocking drug bafilomycin A1 (BafA1) overnight, and fixed and stained for lysosome marker Lamp1. Scale bars, 10 μm. (**C**) TMD mutation strategy. The TM alpha helix of human EMD was divided into N-terminal and

*Figure 2 continued on next page*

*Figure 2 continued*

C-terminal halves, and the aromatic residues in each half were mutated to alanine to generate mutants with similar ΔG-insertion values. For the full TMD mutant, all aromatic residues were mutated to alanine, yielding a predicted ΔG-insertion of –2.299. (**D**) Surface antibody:GFP FACS histogram and (**E**) quantification of the mutants diagrammed in (**C**). N=4 independent experiments. (**F**) Surface antibody:GFP FACS histogram and (**G**) quantification of mouse WT EMD, RA, ΔLEM, and ΔLEMΔQRRR truncation surface expression. N=4 independent experiments. (**H**) Summary of emerin constructs that do or do not traffic. (**I**) Western blot analysis of emerin constructs induced with 2 µg/mL doxycycline for 48 hr, then washed and incubated for an 18 hr chase. (**J**) Quantification of western blot band intensity from (**I**). GFP antibody signal after washout was divided by the respective unwashed condition to yield the fraction GFP remaining after doxycycline washout across three independent replicates. For all panels: *** indicates adjusted p-value<0.0005, * indicates p=0.0104. All p-values were obtained by one-way ANOVA with Šídák's or Tukey's multiple comparisons tests.

The online version of this article includes the following source data and figure supplement(s) for figure 2:

**Source data 1.** Original files for WB displayed in *Figure 2*.

**Source data 2.** PDF files of uncropped WB displayed in *Figure 2*, annotated bands and MW markers.

**Figure supplement 1.** Emerin contains a long, hydrophobic transmembrane domain and a conserved RXR motif.

secretory trafficking, fitting with the hypothesis that low TMD hydrophobicity suffices to stably retain proteins in the ER (*Ronchi et al., 2008*).

Emerin is not the only tail-anchored INM protein with a particularly hydrophobic TMD. Another such protein is LAP2β (*Lomize et al., 2017*), which shares several sequence features with emerin (*Berk et al., 2013*; *Figure 2—figure supplement 1A*, *Figure 3C*). We identified three RXR motifs within the LAP2β N-terminal domain, one of which (designated RXR1) is just downstream of the LEM-like fold – a striking similarity to emerin's RXR motif (*Figure 3C*). To test whether the same localization mechanisms govern LAP2β, we tagged its C-terminus with GFP and expressed it in U2OS cells. LAP2β-GFP was detected at the cell surface, albeit at much lower levels than EMD-GFP (*Figure 3D and E*, *Figure 3—figure supplement 1A and B*). Interestingly, replacing most of the LAP2β N-terminal domain with the soybean ascorbate peroxidase (APEX2) enzyme caused a 20-fold increase in PM expression (*Figure 3D*, *Figure 3—figure supplement 1B*), suggesting that the LAP2β TMD has a similar proclivity for secretory trafficking. Finally, mutating the LAP2β RXR1 to AAA led to a 70% increase in PM accumulation (*Figure 3D*), mirroring the effect of the EMD RA mutation. Together, these data indicate that the emerin and LAP2β TMDs are insufficient for NE/ER targeting and that both proteins contain retention sequences which counteract TMD-dependent secretion.

RXR motifs are thought to bind COPI subunits to return escaped proteins to the ER (*Michelsen et al., 2005*; *Michelsen et al., 2007*), though this mechanism is not understood. To test whether the increased trafficking of the RA mutant results from reduced COPI interaction, we immunoprecipitated N-terminally FLAG-tagged mouse emerin constructs from U2OS cells (*Figure 3—figure supplement 1C*). Mass spectrometry revealed a robust interaction between emerin and the COPI coat, with several COPI subunits ranking among the top hits (*Figure 3F*). Surprisingly, however, this interaction was not weakened by the RXR mutation (*Figure 3G*). We noted that the mouse EMD bait also co-immunoprecipitated endogenous human EMD (*Figure 3G*), opening the possibility that COPI might indirectly interact with both FLAG constructs via endogenous emerin. Alternatively, the RXR motif's function could depend on protein topology. Importantly, while we were able to detect FLAG-EMD in the lysosome by immunofluorescence (IF) (*Figure 3—figure supplement 1D*), FLAG-RA construct was not poorly expressed relative to FLAG-WT (*Figure 3—figure supplement 1C*) in contrast to RA-GFP compared to WT-GFP (*Figure 2I*). We also noted that the lysosomal accumulation of N-terminally FLAG-tagged emerin was more subtle than the C-terminally GFP-tagged version (*Figures 1A and 2B*), indicating that epitope tagging and/or tag orientation influences the extent of emerin protein turnover.

## C-terminal tagging destabilizes emerin via secretory trafficking

To determine whether tagging and/or overexpression increase the extent of emerin trafficking, we needed a system to better control emerin expression. To this end, we generated a human iPSC line with a dual integrase cassette exchange (DICE) landing pad (*Zhu et al., 2014*) in the AAVS1 safe harbor locus (*Figure 4A*), then knocked emerin out of these cells with CRISPR/Cas9 (*Figure 4B*). Integrating GFP-tagged constructs into the parental and EMD knockout (KO) DICE landing pads enabled us to measure EMD-GFP trafficking in the presence or absence of the endogenous copy, respectively (*Figure 4C*). We included an mCherry-P2A upstream of the EMD-GFP to control for minor

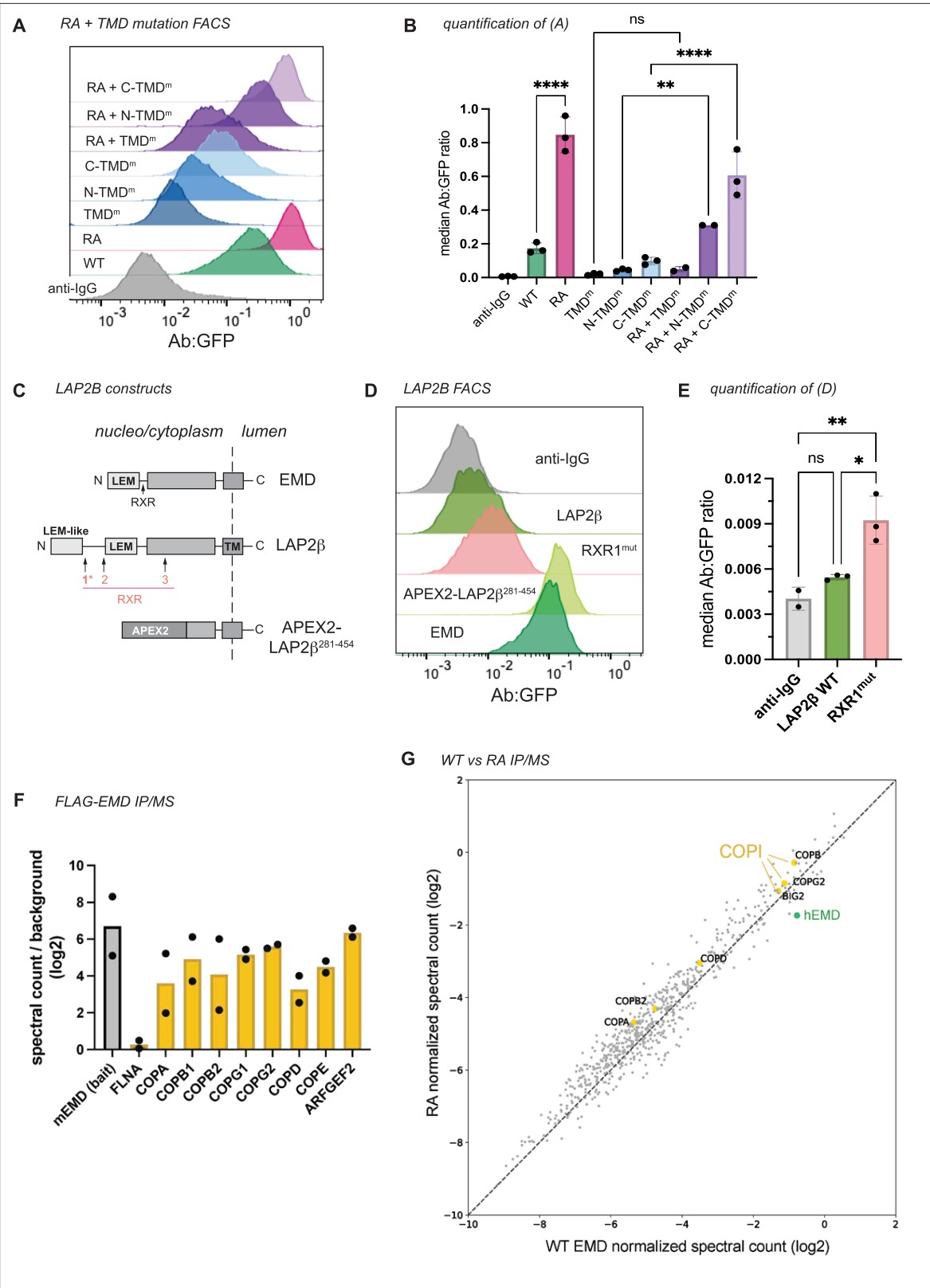

**Figure 3.** RXR motif limits transmembrane domain (TMD)-dependent trafficking of emerin and LAP2β without influencing COPI binding. (**A**) Surface anti-GFP:GFP histogram and (**B**) quantification of indicated RA+ TMD mutant combinations. ** indicates p=0.0021. N=3 independent experiments. (**C**) Diagram of LAP2β domain structure and position of RXR motifs. APEX2 fusion contains no RXR motifs. (**D**) Antibody:GFP histogram and (**E**) quantification of the highest 25% GFP-expressing cells diagrammed in (**C**). RXR^mut: LAP2β RXR1 mutated to AAA. * indicates p=0.0173; ** indicates

*Figure 3 continued on next page*

*Figure 3 continued*

p=0.0072. (**F**) Spectral counts of COPI proteins immunoprecipitated by WT FLAG-EMD normalized to negative control IP. Spectral counts of the mouse EMD bait and common contaminant filamin A (FLNA) plotted for comparison. N=2 independent experiments. (**G**) Spectral counts of proteins immunoprecipitated by FLAG-WT and FLAG-RA mouse EMD normalized to each respective bait. Dotted line represents equal co-immunoprecipitation with the two constructs. COPI proteins and endogenous human EMD highlighted in yellow and green, respectively. All p-values were obtained by one-way ANOVA with Šídák's or Tukey's multiple comparisons tests.

The online version of this article includes the following source data and figure supplement(s) for figure 3:

**Figure supplement 1.** Controls for LAP2B trafficking and emerin immunoprecipitation experiments.

**Figure supplement 1—source data 1.** Original files for WB in *Figure 3—figure supplement 1*.

**Figure supplement 1—source data 2.** PDF files containing uncropped annotated WB for *Figure 3—figure supplement 1*.

differences in construct expression (*Figure 4A*). The GFP-tagged emerin constructs mirrored the trafficking pattern we observed in U2OS cells; the EMD RA-GFP construct accumulated more at the PM, while the TMD mutant did not traffic (*Figure 4D and E*). The steady-state abundance reflected these trafficking patterns. The GFP fluorescence of the TMD mutant accumulated to higher levels, indicating that it was stabilized by its inability to traffic. In contrast, the RA mutant fluorescence was slightly lower, indicating that it was modestly less stable than the WT protein, although this difference is not statistically significant (*Figure 4F and G*). There was no difference in total or surface expression between integrants in the EMD KO and WT DICE backgrounds, suggesting that the trafficking of EMD-GFP is comparable both when moderately overexpressed and when expressed at endogenous levels (*Figure 4D and E*).

Tracking tail-anchored (i.e. untagged/N-terminally tagged) emerin through the secretory pathway is challenging because it lacks a lumenal domain amenable to surface labeling. However, after discovering ways to limit lysosomal flux (via the TMD) or accelerate it (via the RXR motif), we reasoned that the trafficking patterns of tail-anchored mutants could be inferred from their steady-state abundance when expressed from the DICE landing pad. To test whether the GFP tag orientation affects protein expression, we compared the steady-state fluorescence of N- and C-terminally tagged mutants. Surprisingly, the N-terminally tagged (tail-anchored) GFP fusions were four to five times more abundant than the C-terminally tagged (single pass) fusions (*Figure 4F and G*). C-terminal fusions were also more completely relocalized to the lysosome after BafA1 than N-terminal fusions, where NE/ER EMD signal was still apparent (*Figure 4—figure supplement 1A*). Unlike the C-terminal fusions, the N-terminally tagged TMD mutant did not accumulate more than the N-terminally tagged WT protein, suggesting that N-terminally tagged emerin is not destabilized by trafficking. We interpret this finding with caution, as we observed a slight decrease in the level of the N-terminally tagged TMD mutant compared to WT (*Figure 4G*). Since the insertion of tail-anchored proteins relies on TMD hydrophobicity (*Rapaport and Herrmann, 2023*; *Shao and Hegde, 2011*), improper targeting to the ER membrane due to TMD mutation would confound our conclusion that N-terminally tagged constructs undergo similarly low levels of trafficking. Interestingly, upon treatment with BafA1, the N-terminally tagged WT and RA constructs formed GFP+ lysosomal puncta, while the N-terminally tagged TMD mutant did not (*Figure 4—figure supplement 1A*). This suggests that N-terminally tagged emerin does undergo some level of TMD-dependent trafficking. Further, the N-terminal RA fusion was slightly less fluorescent at steady state than WT (*Figure 4G*), indicating that the RXR motif also stabilizes tail-anchored GFP-EMD.

Because of the profound differences in the distribution of N- and C-terminally tagged constructs, we investigated how lumenal GFP affects emerin's early secretory trafficking by analyzing its enrichment in ER exit sites (ERES). ER cargoes that are actively packaged into ERES can be distinguished from bulk flow secretory cargoes by incubating cells at 10°C, where ERES form and cargoes accumulate but cannot progress toward the Golgi (*Mezzacasa and Helenius, 2002*). We compared the co-localization of endogenous untagged emerin, WT-GFP, and TMD$^m$-GFP with the ERES marker SEC31 at 37°C and 10°C (*Figure 4—figure supplement 1B and C*). We found that GFP-tagged emerin co-localized with SEC31 at steady state, while this was not the case for endogenous emerin or the tagged TMD mutant. ERES co-localization was not increased at 10°C for any of the emerin variants analyzed, indicating that emerin does not contain specific ER exit signals.

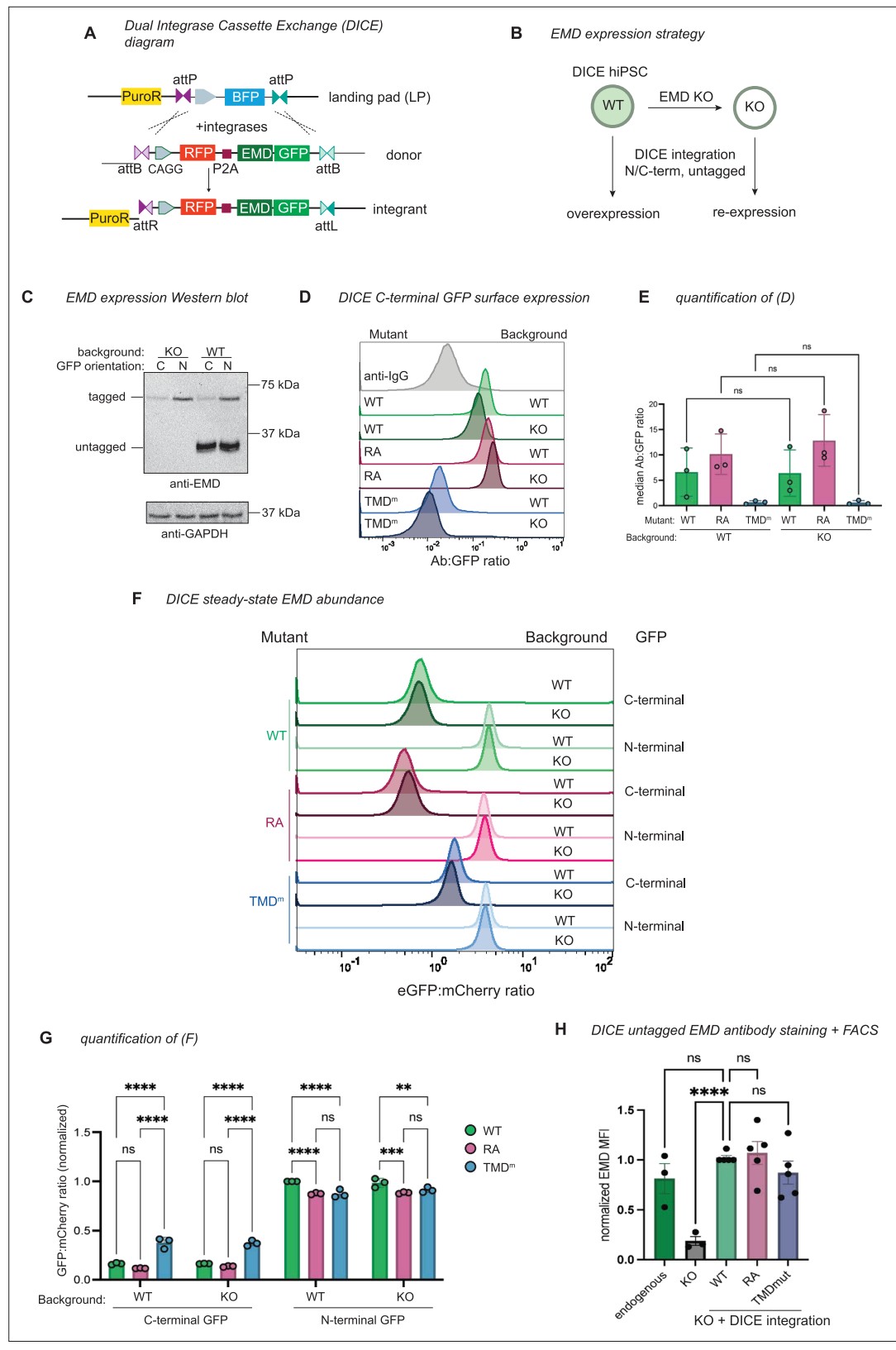

**Figure 4.** Safe harbor expression reveals that C-terminal GFP destabilizes emerin. (**A**) Diagram of emerin integration into the AAVS1 locus. Landing pad BFP is exchanged for mCherry-P2A-emerin±GFP via Bxb1 and PhiC31 integrases. Integrases irreversibly recombine landing pad attP and donor attB sites into attR and attL sites. (**B**) Strategy to compare overexpressed emerin to knockout (KO) rescue. (**C**) Western blot analysis of WT and

*Figure 4 continued on next page*

*Figure 4 continued*

EMD KO dual integrase cassette exchange (DICE) hiPSCs expressing N- and C-terminally GFP-tagged emerin. Untagged endogenous EMD and GFP-tagged EMD are detected by the same anti-EMD antibody. (**D**) Anti-GFP surface labeling histogram and (**E**) quantification of C-terminally tagged emerin integrants from (**B**–**C**). Statistical significance was determined using one-way ANOVA with Šídák's multiple comparisons test. N=3 independent experiments. (**F**) FACS plot and (**G**) quantification of steady-state GFP abundance. Data were normalized to the GFP-EMD signal in the EMD WT background. N=3 independent experiments. ** indicates p=0.0079 by two-way ANOVA with Tukey's multiple comparisons test. (**H**) DICE landing pad WT, EMD KO, and KO re-integrated with untagged constructs were lifted, fixed, and stained with anti-EMD antibody. Fluorescence was quantified by flow cytometry and normalized to the WT DICE integrant. MFI, median fluorescence intensity of antibody signal. N=5 replicates over 4 independent experiments. **** indicates p<0.0001 by mixed effects analysis.

The online version of this article includes the following source data and figure supplement(s) for figure 4:

**Source data 1.** Original files for WB displayed in *Figure 4*.

**Source data 2.** PDF file containing uncropped annotated WB in *Figure 4*.

**Figure supplement 1.** Differential trafficking behavior of tagged and untagged emerin.

Collectively, these data indicate that N- and C-terminally tagged emerin constructs travel to the lysosome via their hydrophobic TMDs, although C-terminal tags accelerate this trafficking by encouraging bulk flow ER exit.

Finally, we integrated untagged WT, RA, and TMD mutant constructs into EMD KO landing pad cells, then fixed and stained cells for confocal microscopy and flow cytometry. We saw no apparent differences in the localization or intensity of any of the emerin mutants, though the data were considerably noisier than the steady-state GFP measurements (*Figure 4H*). We were also unable to detect untagged emerin in the lysosome after BafA1 treatment (*Figure 4—figure supplement 1D*), suggesting it does not traffic through the secretory pathway. However, the sensitivity of these assays may be limited by fixation, permeabilization, and antibody staining; therefore, we cannot rule out the possibility that small differences do exist between untagged emerin variants.

## Discussion

In this work, we dissected how exogenous epitope tags and intrinsic sequence elements influence the secretory trafficking of the INM-resident protein emerin. We discovered that the aromatic amino acids in emerin's 23-residue TMD facilitate its travel to the lysosome. In the absence of other signals, TMD length and hydrophobicity are sufficient to partition tail-anchored proteins along the secretory pathway (*Ronchi et al., 2008*; *Borgese, 2016*). The biophysical properties of the emerin and LAP2β TMDs resemble PM, rather than ER, resident TMDs (*Figure 2—figure supplement 1*, *Sharpe et al., 2010*). However, this notable hydrophobicity is likely important for the proteins' biogenesis. Tail-anchored proteins with very hydrophobic TMDs are posttranslationally targeted by the TRC40/GET complex, which captures C-terminal alpha helices after they emerge from the ribosome and delivers them to the ER membrane (*Rapaport and Herrmann, 2023*). Emerin and Lap2β are established TRC40 client proteins (*Coy-Vergara et al., 2019*; *Pfaff et al., 2016*), so their hydrophobic TMDs are likely required for TRC40 recognition. However, adding a C-terminal GFP converts a tail-anchored protein to a single-pass transmembrane protein, which will instead be co-translationally targeted to the ER by the SRP pathway (*Shao and Hegde, 2011*; *Walter et al., 1981*). Our results indicate that this change in topology displaces emerin, making C-terminally tagged emerin less stable and more prone to lysosomal trafficking than its N-terminally tagged counterpart (*Figure 4F and G*, *Figure 4—figure supplement 1A*). Consistently, we found that only lumenally tagged emerin sampled ERES, and that this was reduced by manipulating the TMD hydrophobicity (*Figure 4—figure supplement 1B and C*). This suggests that hydrophobic TMDs are less energetically favored within the ER membrane when attached to a lumenal domain, or that different biogenesis pathways bias cargoes toward ER exit.

We hypothesize that active ER retention plays a role in INM protein targeting. In interphase, emerin is not entirely sequestered at the INM by the nuclear lamina; a pool of emerin protein remains in the peripheral ER (*Nastały et al., 2020*; *Salpingidou et al., 2007*). The intrinsic ability of emerin's TMD to traffic from the ER implies that a limiting mechanism must exist to bias the protein toward diffusion to the INM. Emerin could stably reside in the ER in part by binding to other ER-resident proteins.

Interestingly, emerin was reported to interact with the ER tethering protein MOSPD3 via its YEESY (95-99) motif (*Cabukusta et al., 2020*), which we previously found to limit trafficking (*Buchwalter et al., 2019*). While both TMDs traffic to the PM, full-length LAP2β-GFP is more retained in the NE/ER than EMD-GFP (*Figure 3D*, *Figure 3—figure supplement 1A and B*), indicating that the N-terminal domains limit trafficking to different extents.

We identified the RXR motif as a shared negative regulator of TMD-dependent trafficking, prompting us to revise our original model: the exposure of this motif, not the absence of the LEM domain, prevents ER exit of EMD-GFP (*Figure 2F and G*). This suggests that the sequences around the RXR motif influence its activity by steric occlusion or other mechanisms. RXR motifs prevent the ER exit of some PM-resident proteins until they are masked by oligomerization, thus coupling ER exit to folding and/or assembly (*Michelsen et al., 2005*). Like the canonical ER retrieval motif KKXX, RXR motifs are thought to engage COPI subunits to return escaped ER membrane proteins from the early Golgi (*Michelsen et al., 2005*; *Marcello et al., 2010*). Yeast-2-hybrid experiments predicted RXR signals to interact with the beta and delta subunits of the COPI coat (*Michelsen et al., 2007*), though the molecular details are still unclear. Contrary to KKXX motifs, found only on C-termini, RXR motifs may be on N-termini or within a polypeptide chain (*Michelsen et al., 2005*). Truncated, lumenally tagged variants of Sun2, another tail-anchored INM protein, require a poly-arginine motif to prevent mislocalization to the Golgi (*Turgay et al., 2010*). *Turgay et al., 2010*, observed purified Sun2 RXR mutants losing interaction with COPI subunits in GST pull-down assays. In contrast, we did not observe loss of COPI binding in our RXR mutant immunoprecipitations. This discrepancy may be explained by differences in methods (purified recombinant protein binding assays vs native IP) or by oligomerization of tagged emerin with endogenous emerin to indirectly bind COPI in lysate (*Fernandez et al., 2021*). Emerin's interaction with the COPI coat was recently detected by proximity biotinylation in mouse P19 cells (*Zhao et al., 2021*). Interestingly, emerin was the only non-Golgi-resident protein identified in the gamma-COP subunit interactome, indicating that endogenous emerin might be a COPI cargo. While we predict this interaction occurs via emerin's RXR motif, we cannot exclude the possibility of other COPI-interacting mechanisms.

Our previous work identified EMD as an unstable protein subject to significant lysosomal flux (*Buchwalter et al., 2019*). However, we have now discovered that emerin's trafficking is markedly promoted by C-terminal GFP tagging, likely influencing many of our initial observations. While we used C-terminal tags to avoid disrupting emerin's N-terminal interactions with chromatin and lamins, this approach inadvertently altered its localization. We did detect trafficking of N-terminally tagged emerin but found that it occurs at a much lower rate. We conclude that hydrophobic tail-anchored TMDs are readily dislocated by C-terminal tags and echo the warnings raised by generations of cell biologists regarding tag-induced localization artifacts (*Stadler et al., 2013*; *Montecinos-Franjola et al., 2020*; *Margolin, 2012*). We did not detect trafficking of the untagged emerin constructs. Nonetheless, because trafficking is detectable at or below endogenous levels with different tags and orientations, we cannot rule out the possibility that emerin traffics under specific conditions. We note poor correlation of emerin transcript and protein across tissues (*Wang et al., 2019*; *Figure 4—figure supplement 1E*), which suggests protein-level regulation that may dynamically regulate emerin levels. While we still do not know whether emerin's ability to travel beyond the NE/ER influences its function, our experiments reveal an unanticipated connection between the topology and localization of INM proteins and raise new questions about the balance between secretion and retention of INM-destined proteins.

## Methods
### Cell culture and cell line generation

U2OS cells were obtained from and verified by the UCSF Cell and Genome Engineering Core. Cells were cultured in McCoy's 5A medium supplemented with 10% FBS and penicillin/streptomycin. To generate EMD-GFP cell lines, U2OS were seeded into six-well plates at a density of 300,000 cells per well. The next day, a transfection mix of 0.5 µg EMD-GFP donor plasmid, 0.25 µg PB200A-1 PiggyBac Transposase plasmid, 3.5 µL Lipofectamine 2000, and 300 µL Opti-MEM was added to each well. After 24 hr, cells were split 1:2 and selected in 5 µg/mL blasticidin for 1 week, then maintained in 5 µg/mL blasticidin thereafter. Prior to an experiment, cells were split into media without blasticidin

and induced with 1 μg/mL doxycycline for 24–48 hr. To generate FLAG-EMD cell lines, 2 μg CMV-FLAG-EMD plasmid was transfected without PiggyBac Transposase. Cells were selected with 400 μg/mL G418 for 1 week, then maintained in 200 μg/mL G418.

WTC-11 hiPSCs were obtained from and verified by the Berkeley Stem Cell Center. Cells were maintained on Matrigel-coated vessels in mTeSR+ without penicillin/streptomycin. Cells were passaged as clumps (using ReLeSR) for routine culture and as single cells (using Accutase) for counting and electroporating. Cells were plated in ROCK inhibitor Y-27632, then changed to fresh mTeSR+ after 24 hr. CRISPR editing was performed by electroporating 12.5 pmol pure Cas9-NLS and 12.5 pmol sgRNA into 1 million cells using a 100 μL Neon electroporator set to 1300 V, 30 ms, and 1 pulse. For DICE landing pad installation, we included 2 μg cassette-containing HDR template plasmid in the transfection mix. Landing pad cells were selected in 0.25 μg/mL puromycin for 3 days. Single clones were isolated by limiting dilution in 96-well plates, then expanded and genotyped for heterozygous insertion using primers spanning the AAVS1 homology arms. Emerin KOs were generated from one landing pad clone using two sgRNAs targeting the start and end of the coding sequence. Clones were genotyped by PCR to verify the entire region was deleted, then by western blot to confirm the loss of protein. To integrate emerin constructs into the DICE landing pad, we electroporated 1 million cells with 2 μg attB-containing donor plasmid and 500 ng BxbI-P2A-PhiC31 integrase plasmid. After letting cells recover, we isolated landing pad BFP–/donor+ cells using a SONY SH800 cell sorter. Cells were checked for mycoplasma every 3–4 months and were found to be negative.

## Plasmid cloning

All U2OS constructs except FLAG-EMD were expressed in the XLone all-in-one Tet-ON Piggybac plasmid (*Randolph et al., 2017*). Mouse emerin sequences were cloned into XLone from plasmids used in our previous study. Human emerin and LAP2β were amplified from Hek293T and WTC-11 cDNA using primers with overhangs to XLone. Mouse cytochrome B5 was amplified from E14 mESC cDNA. cDNA was synthesized using NEB MMLV reverse transcriptase. FLAG-EMD constructs were expressed from the CMV promoter in a Neo/Kan-resistant vector.

## Emerin constructs

| Designation | Gene | Species | Uniprot ID | Mutation |
|---|---|---|---|---|
| LAP2β RXR1 | LAP2β | Human | P42167 | 48–50 →AAA |
| LAP2β APEX2-TMD | LAP2β | Human | P42167 | 1–280 →APEX2 |
| hEMD RA | EMD | Human | P50402 | 44–47 →AAAA |
| hEMD ΔLEM | | Human Mouse | P50402 O08579 | D1–43 |
| hEMD ΔLEM ΔQRRR | EMD | Human Mouse | P50402 O08579 | D1–47 |
| hEMD Cyb5 TM | EMD | Human | P50402 (EMD) P56395 (Cyb5) | EMD 1–224+mouse cyb5a 95–134 |
| mEMD Cyb5 TM | EMD | Mouse | O08579 (EMD) P56395 (Cyb5) | EMD 1–225+mouse Cyb5a 95–134 |
| TMD-GFP | EMD | Mouse | O08579 | 212–259 |
| N-TMD^m | EMD | Human | P50402 | W226A, F232A, F235A |
| C-TMD^m | EMD | Human | P50402 | F240A, F241A, Y243A |
| Full TMD^m | EMD | Human | P50402 | W226A,F232A, F235A, F240A, F241A, Y243A |

Small emerin mutations were generated using the NEB site-directed mutagenesis kit, while larger fusions were assembled using the NEB HiFi Assembly kit.

The DICE landing pad cassette contained a PGK promoter driving HSV thymidine kinase-P2A-TagBFP, all flanked by BxbI and PhiC31 attP sites. This cassette was built into an AAVS1 targeting vector (a generous gift from Dr. Tom Nowakowski) containing a splice acceptor-P2A-puro. DICE

donor constructs contained Bxb1 and PhiC31 attB sites flanking the CAGGS promoter and an mCherry-P2A-EMD.

Plasmids were prepped using the ZymoPURE II midiprep kit. All plasmids were verified by Sanger or whole plasmid nanopore sequencing before transfection.

## TMD hydrophobicity calculations

Transmembrane segments were obtained for EMD, LAP2β, STX3, and CYB5 from Uniprot annotations. DG-ins was calculated using the DGprediction algorithm (*Hessa et al., 2007*). Median DG-transfer and TM length values for secretory membranes were obtained from the human Membranome database (*Lomize et al., 2017*; *Lomize et al., 2018*). Data obtained from the Membranome database were plotted in GraphPad Prism.

## Surface labeling and flow cytometry

The surface staining assay was adapted from *Welch et al., 2021*. Briefly, cells were washed with PBS, lifted for 3–5 min with accutase, then quenched with cold complete media. Cells were transferred to microfuge tubes and placed on ice for the duration of staining and washing. Cells were pelleted 800×$g$ for 3 min at 4°C, then washed once in 2% FBS/PBS. Cells were stained in 80 µL anti-GFP 647 (1:20 in 2% FBS/PBS) or anti-rabbit IgG Alexa Fluor 647 secondary antibody (1:200 in 2% FBS/PBS) as a negative control. The cells used for the anti-IgG controls were WT EMD-GFP (for all EMD comparisons) or WT LAP2β-GFP (for LAP2β comparisons). After staining for 30 min, cells were diluted with 200 µL 2% FBS/PBS, pelleted and washed twice with 400 µL 2% FBS/PBS, and then optionally fixed in 4% paraformaldehyde (PFA/PBS) for 10 min at room temperature. Fixed cells were washed twice and then stored at 4°C until being resuspended in PBS and strained into round-bottom FACS tubes before analysis.

For flow analysis of antibody-stained proteins, cells were lifted with accutase, washed with PBS, and fixed by rotating in 4% PFA/PBS for 10 min. After washing twice in PBS, cells were permeabilized in IF buffer (0.1% Triton X-100, 0.02% SDS, 10 mg/mL BSA in PBS) for 30 min before incubating in primary antibody solution for 1–2 hr at room temperature. Cells were washed three times in IF buffer, incubated with fluorescent secondary antibodies for 1 hr, and then washed three more times before being strained into FACS tubes.

Cells were analyzed on a BD FACSVerse or a Thermo Fisher Attune NxT. At least 50,000 cells were analyzed per sample. Flow cytometry data were analyzed in FlowJo. Surface antibody:GFP ratio was obtained using the 'derive parameters' function and applied to the GFP+ populations. Histograms are normalized to the mode of each sample for visualization.

## Immunofluorescence microscopy

Cells were seeded in Ibidi culture chambers (Matrigel-coated for hiPSCs) prior to an IF experiment. For lysosome co-localization analysis, cells were treated with 100 nM bafilomycin A1 for 16–18 hr. Cells were washed in PBS, then fixed in 4% PFA in PBS for 5 min at room temperature. For LAMP1 staining, cells were permeabilized in 0.1% digitonin/PBS at 4°C for 10 min, then blocked in 2% FBS/ PBS. LAMP1 antibody was diluted 1:200 in 2% FBS/PBS. For emerin and FLAG immunostaining, fixed cells were permeabilized in IF buffer for 30 min, then incubated in primary antibodies (diluted 1:250 and 1:1000 in IF buffer, respectively) for 1 hr at room temperature. After washing, cells were incubated with Alexa Fluor-conjugated secondary antibodies and Hoechst 33342 (both 1:1000) for 30 min at room temperature. Emerin/FLAG staining is incompatible with LAMP1 staining because of differential permeabilization requirements. In these cases, we had to rely on the formation of EMD+ puncta after bafilomycin treatment to infer lysosome localization.

Selective permeabilization and ERES co-localization temperature blocks were performed as previously described (*Vander Heyden et al., 2011*). For selective permeabilization topology analysis, U2OS cells were induced with doxycycline for 24 hr, washed with cold PBS, and incubated with cold 0.0025% digitonin/PBS or PBS only for 5 min at 4°C. Cells were then fixed with 4% PFA for 5 min at room temperature. To permeabilize all endomembranes, cells were incubated with IF buffer (0.1% Triton X-100, 0.02% SDS, 10 mg/mL) BSA in PBS for 30 min, then stained with anti-EMD and anti-GFP antibodies diluted in 2% FBS/PBS.

For ERES co-localization, EMD-GFP U2OS cells were induced with doxycycline for 4 hr or left unin-duced to analyze newly synthesized or endogenous emerin, respectively. Cells were then placed on a metal platform submerged in a 10°C water bath or continued incubation at 37°C for 2 hr, then fixed and stained for endogenous emerin and Sec31a to mark ERES.

Images were acquired using a Nikon CSU-X1 spinning disk confocal microscope with ×40/1.3 NA or ×60/1.4 NA oil objectives. 20–30 Z-slices were imaged per cell with a step size of 0.3 μm. 16-bit images were saved as ND2 files using the Nikon Elements 5.02 build 1266 software. Representative slices were selected and cropped in Fiji. Pearson correlation was calculated using the Fiji PSC co-lo-calization plug-in.

## Protein stability measurement and western blotting

For doxycycline washout experiments, 300,000 U2OS cells were seeded in 12-well plates with 2 μg/mL doxycycline. After 48 hr, half the wells were washed twice with PBS and incubated in fresh medium without doxycycline for 18 hr. Cells were washed twice with PBS and lysed in the wells by scraping in 80 μL RIPA buffer (50 mM Tris, 150 mM NaCl, 1% Triton X-100, 0.5% deoxycholate, 0.1% SDS) plus protease inhibitors and benzonase. Lysates were spun 17,000×$g$ at 4°C for 10 min, then supernatants were mixed with sample buffer and boiled for 10 min. 12 μL of each sample was loaded onto home-made 10% SDS-PAGE gels, then transferred onto nitrocellulose membranes. Blots were incubated at 4°C overnight in primary antibody diluted with 5% milk/TBST, then washed and incubated in HRP-conjugated secondary antibodies for 1 hr at room temperature. Blots were developed using Pierce ECL Western Blotting Substrate and imaged using the GelDoc Imaging System. Band intensity was quantified using Fiji.

## FLAG-EMD immunoprecipitation and mass spectrometry

FLAG-EMD, FLAG-RA, and parental U2OS (non-expressing negative control) cells were grown to confluence in three 10 cm plates per cell line, then harvested by scraping and transferring to a micro-fuge tube. Cells were pelleted and lysed in 1 mL lysis buffer (50 mM Tris-HCl pH 7.4, 150 mM NaCl, 0.3% Triton X-100, 1 mM DTT, 2 mM EDTA, 5 mM MgCl$_2$, 1x protease inhibitor cocktail, 1x PhosSTOP, benzonase), then incubated for 30 min end over end at 4°C. Lysates were probe-sonicated in six rounds of 10 s pulses at 20% power, then spun 17,000×$g$ at 4°C for 25 min. The supernatant was added to 40 μL FLAG M2 magnetic beads which had been calibrated in 500 μL lysis buffer and sepa-rated on a magnetic stand. Samples were incubated end over end at room temperature for 1 hr, then magnetically separated and washed three times with 1 mL wash buffer (100 mM HEPES-KOH pH 7.5, 100 mM KCl, 5% glycerol, 0.1% NP-40, protease inhibitors). After the final wash, the supernatant was discarded, and the beads were resuspended in 200 μL elution buffer (100 μg/mL 3× FLAG peptide in wash buffer) and eluted for 1 hr at room temperature. Eluates were precipitated with four volumes of ice-cold acetone overnight at –20°C, spun at maximum speed for 5 min, then air-dried. Pellets were processed for mass spectrometry using a PreOmics iST kit according to the manufacturer's instructions.

## Mass spectrometry and analysis

A nanoElute was attached in line to a timsTOF Pro equipped with a CaptiveSpray Source (Bruker, Hamburg, Germany). Chromatography was conducted at 40°C through a 25 cm reversed-phase C18 column (PepSep) at a constant flow rate of 0.5 μL/min. Mobile phase A was 98/2/0.1% water/MeCN/formic acid (vol/vol/vol) and phase B was MeCN with 0.1% formic acid (vol/vol). During a 108 min method, peptides were separated by a three-step linear gradient (5–30% B over 90 min, 30–35% B over 10 min, 35–95% B over 4 min) followed by a 4 min isocratic flush at 95% for 4 min before washing and a return to low organic conditions. Experiments were run as data-dependent acquisitions with ion mobility activated in PASEF mode. MS and MS/MS spectra were collected with $m/z$ 100–1700 and ions with z = +1 were excluded. Raw data files are available on the Mass Spectrometry Interactive Virtual Environment (MassIVE), a full member of the Proteome Xchange consortium under the identi-fier: MSV000096858.

Raw data files were searched using PEAKS Online Xpro 1.6 (Bioinformatics Solutions Inc, Waterloo, Ontario, Canada). The precursor mass error tolerance and fragment mass error tolerance were set to 20 ppm and 0.03, respectively. The trypsin digest mode was set to semi-specific, and missed cleavages were set to 2. The human Swiss-Prot reviewed (canonical) database (downloaded from

UniProt) totaling 20,385 entries or a custom database using human and mouse emerin (downloaded from uniprot, EMD_human P50402, EMD_mouse O08579) was used. Carbamidomethylation (C) was selected as a fixed modification. Oxidation (M) was selected as a variable modification.

Resulting datasets were subjected to the following filtration criteria: (1) Database search ($-10 \log$(p-value)$\geq$20, 1% peptide and protein FDR). (2) Identified proteins with less than 3% coverage, three unique peptides, or spectral areas less than 5000 were discarded. (4) Spectral counts for each protein were normalized to corresponding counts from the negative control IP, and then to the protein length. (5) The ratio of normalized counts to the mouse emerin bait was plotted to obtain relative co-immunoprecipitation efficiency.

## Acknowledgements

We are grateful to UCSF core facility staff for training and technical support. The Gladstone Institutes Stem Cell Core provided hiPSC culture training and equipment. Images were acquired at the Center for Advanced Light Microscopy – CVRI Microscopy core on microscopes purchased through the UCSF Research Evaluation and Allocation Committee, the Gross Fund, and the Heart Anonymous Fund. Flow cytometers and cell sorters were purchased and supported by the UCSF Center for Live Cell Analysis core. We also thank Elphege Nora for guidance with integrase cassette exchange, Emmy Delaney for experimental assistance, and Tracy Knight for experimental assistance and feedback on this manuscript. Adam Frost, Shaeri Mukherjee, Peter Walter, Martin Kampmann, Willow Coyote-Maestas, and members of the Buchwalter lab provided valuable perspective and discussion throughout this work. AB was supported by the Chan Zuckerberg Biohub, and JM was supported by the National Heart, Lung, and Blood Institute (F31HL170757).

## Additional information

### Funding

| Funder | Grant reference number | Author |
| --- | --- | --- |
| Chan Zuckerberg Initiative | | Abigail Buchwalter |
| National Heart and Lung Institute | F31HL170757 | Jessica Mella |

The funders had no role in study design, data collection and interpretation, or the decision to submit the work for publication.

### Author contributions

Jessica Mella, Conceptualization, Data curation, Formal analysis, Funding acquisition, Validation, Investigation, Methodology, Writing – original draft, Writing – review and editing; Regan F Volk, Formal analysis, Investigation, Methodology; Balyn W Zaro, Resources, Formal analysis, Investigation, Methodology; Abigail Buchwalter, Conceptualization, Resources, Formal analysis, Supervision, Funding acquisition, Investigation, Visualization, Methodology, Project administration, Writing – review and editing

### Author ORCIDs

Jessica Mella ⓘ https://orcid.org/0000-0001-7541-6789
Regan F Volk ⓘ https://orcid.org/0000-0001-6748-719X
Balyn W Zaro ⓘ https://orcid.org/0000-0002-8938-9889
Abigail Buchwalter ⓘ https://orcid.org/0000-0001-7181-6961

Reviewer #1 (Public review): https://doi.org/10.7554/eLife.105937.3.sa1
Reviewer #2 (Public review): https://doi.org/10.7554/eLife.105937.3.sa2
Author response https://doi.org/10.7554/eLife.105937.3.sa3

## Data availability

Mass spectrometry data referenced in Figure 3 have been deposited in the MassIVE database under accession number MSV000096858.

The following dataset was generated:

| Author(s) | Year | Dataset title | Dataset URL | Database and Identifier |
|---|---|---|---|---|
| Buchwalter A | 2025 | C-terminal tagging, transmembrane domain hydrophobicity, and an ER retention motif influence the secretory trafficking of the inner nuclear membrane protein emerin | https://massive.ucsd.edu/ProteoSAFe/dataset.jsp?task=a3eba071fc934bab8a69dd9c2149f9f2 | MassIVE, MSV000096858 |

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

# Appendix 1

## Appendix 1—key resources table

| Reagent type (species) or resource | Designation | Source or reference | Identifiers | Additional information |
|---|---|---|---|---|
| Gene (*Homo sapiens*) | EMD | NA | NCBI RefSeq:NM_000117.3 | |
| Gene (*Homo sapiens*) | Lap2B | NA | NCBI RefSeq:NM_001032283.3 | |
| Gene (*Mus musculus*) | EMD | NA | NCBI RefSeq:NM_007927.4 | |
| Gene (*Mus musculus*) | Cyb5a | NA | NCBI RefSeq:NM_025797.4 | |
| Cell line (human, female) | U2OS | UCSF Cell and Genome Engineering Core | ATCC:HTB-96 | |
| Cell line (human, male) | WTC-11 hiPSC | Berkeley Stem Cell Center | hPSCreg:UCSFi001-A | |
| Cell line (human, male) | AAVS1 DICE landing pad hiPSC+EMD KO | This paper | | WTC-11 with single-copy landing pad at AAVS1 locus, EMD gene deletion |
| Cell line (human, male) | AAVS1 DICE landing pad hiPSC | This paper | | WTC-11 with single-copy landing pad at AAVS1 locus |
| Transfected construct (human, DICE landing pad hiPSC) | EMD-GFP DICE donor | This paper | Uniprot ID: P50402; pJM358 | Stable single-copy integration of EMD WT (C-terminal tag) |
| Transfected construct (human, DICE landing pad hiPSC) | GFP-EMD DICE donor | This paper | Uniprot ID: P50402; pJM417 | Stable single-copy integration of EMD WT (N-terminal tag) |
| Transfected construct (human, DICE landing pad hiPSC) | RA-GFP DICE donor | This paper | Uniprot ID: P50402; pJM415 | Stable single-copy integration of EMD mutant residues 44–47→AAAA (C-terminal tag) |
| Transfected construct (human, DICE landing pad hiPSC) | GFP-RA DICE donor | This paper | Uniprot ID: P50402; pJM418 | Stable single-copy integration of EMD mutant residues 44–47→AAAA (N-terminal tag) |
| Transfected construct (human, DICE landing pad hiPSC) | TMDm-GFP DICE donor | This paper | Uniprot ID: P50402; pJM416 | Stable single-copy integration of EMD mutant W226A, F232A, F235A, F240A, F241A, Y243A (C-terminal tag) |
| Transfected construct (human, DICE landing pad hiPSC) | GFP-TMDm DICE donor | This paper | Uniprot ID: P50402; pJM419 | Stable single-copy integration of EMD mutant W226A, F232A, F235A, F240A, F241A, Y243A (N-terminal tag) |
| Transfected construct (human, DICE landing pad hiPSC) | Untagged EMD DICE donor | This paper | Uniprot ID: P50402; pJM401 | Stable single-copy integration of EMD WT |
| Transfected construct (human, DICE landing pad hiPSC) | Untagged RA DICE donor | This paper | Uniprot ID: P50402; pJM411 | Stable single-copy integration of EMD mutant |
| Transfected construct (human, DICE landing pad hiPSC) | Untagged TMDm DICE donor | This paper | Uniprot ID: P50402; pJM412 | Stable single-copy integration of EMD mutant W226A, F232A, F235A, F240A, F241A, Y243A |
| Transfected construct (human, U2OS) | XLone EMD-GFP | This paper | Uniprot ID: P50402; pJM225 | PiggyBac vector for stable expression of hEMD WT |
| Transfected construct (human, U2OS) | XLone RA-GFP | This paper | Uniprot ID: P50402; pJM271 | PiggyBac vector for stable expression of EMD 47–50→AAAA |
| Transfected construct (human, U2OS) | XLone ΔLEM-GFP | This paper | Uniprot ID: P50402; pJM272 | PiggyBac vector for stable expression of EMD Δ1–43 |
| Transfected construct (mouse, U2OS) | XLone TMD-GFP | This paper | Uniprot ID: P50402; pJM217 | PiggyBac vector for stable expression of mEMD 212–259 |
| Transfected construct (mouse, U2OS) | XLone GFP-TMD | This paper | Uniprot ID: P50402; pJM222 | PiggyBac vector for stable expression of mEMD 212–259 |
| Transfected construct (human, U2OS) | XLone N-TMDm-GFP | This paper | Uniprot ID: P50402; pJM328 | PiggyBac vector for stable expression of EMD W226A, F232A, F235A |

*Appendix 1 Continued on next page*

*Appendix 1 Continued*

| Reagent type (species) or resource | Designation | Source or reference | Identifiers | Additional information |
| --- | --- | --- | --- | --- |
| Transfected construct (human, U2OS) | XLone C-TMDm-GFP | This paper | Uniprot ID: P50402; pJM330 | PiggyBac vector for stable expression of EMD F240A, F241A, Y243A |
| Transfected construct (human, U2OS) | XLone TMDm-GFP | This paper | Uniprot ID: P50402; pJM331 | PiggyBac vector for stable expression of EMD W226A, F232A, F235A, F240A, F241A, Y243A |
| Transfected construct (human, U2OS) | XLone N-TMDm-GFP+RA | This paper | Uniprot ID: P50402; pJM355 | PiggyBac vector for stable expression of EMD W226A, F232A, F235A, and 47-50AAAA |
| Transfected construct (human, U2OS) | XLone C-TMDm-GFP+RA | This paper | Uniprot ID: P50402; pJM356 | PiggyBac vector for stable expression of EMD F240A, F241A, Y243A, and 47-50AAAA |
| Transfected construct (human, U2OS) | XLone TMDm-GFP+RA | This paper | Uniprot ID: P50402; pJM357 | PiggyBac vector for stable expression of EMD W226A, F232A, F235A, F240A, F241A, Y243A, and 47-50AAAA |
| Transfected construct (human, U2OS) | XLone EMD+Cyb5 TMD-GFP | This paper | Uniprot ID: P50402; pJM273 | PiggyBac vector for stable expression |
| Transfected construct (human, U2OS) | XLone LAP2B-GFP | This paper | Uniprot ID: P42167; pJM326 | PiggyBac vector for stable expression of LAP2B WT |
| Transfected construct (human, U2OS) | XLone LAP2B RxR1-GFP | This paper | Uniprot ID: P42167; pJM333 | PiggyBac vector for stable expression of LAP2B 48–50 →AAA |
| Transfected construct (human, U2OS) | XLone APEX2-LAP2B TMD-GFP | This paper | Uniprot ID: P42167 (LAP2B); pJM312 | PiggyBac vector for stable expression of APEX2+LAP2B 281–454 |
| Transfected construct (mouse, U2OS) | XLone EMD-GFP | This paper | Uniprot ID: O08579; AB_E00175 | PiggyBac vector for stable expression of mEMD WT |
| Transfected construct (mouse, U2OS) | XLone RA-GFP | This paper | Uniprot ID: O08579; pJM228 | PiggyBac vector for stable expression of mEMD 47–50 →AAAA |
| Transfected construct (mouse, U2OS) | XLone DLEM-GFP | This paper | Uniprot ID: O08579; pJM202 | PiggyBac vector for stable expression of mEMD delta 1–43 |
| Transfected construct (mouse, U2OS) | XLone EMD+Cyb5 TMD-GFP | This paper | Uniprot ID: O08579 (EMD); Uniprot ID:P56395 (Cyb5); pJM214 | PiggyBac vector for stable expression of mEMD 1–225+Cyb5 95–134 |
| Transfected construct (mouse, U2OS) | XLone DLEMDQRRR-GFP | This paper | Uniprot ID: O08579; pJM227 | PiggyBac vector for stable expression of mEMD delta 1–47 |
| Transfected construct (mouse, U2OS) | CMV-FLAG-EMD | This paper | Uniprot ID: O08579; pJM244 | Constitutive stable expression of FLAG-EMD |
| Transfected construct (mouse, U2OS) | CMV-FLAG-RA | This paper | Uniprot ID: O08579; pJM258 | Constitutive stable expression of FLAG-mEMD 47-50AAAA |
| Antibody | Anti-EMD rabbit polyclonal | ProteinTech | 10351-1-AP | WB: 1:2000, IF: 1:250 |
| Antibody | Rabbit polyclonal anti-GFP | ChromoTek | PABG1 | IF: 1:1000 for selective permeabilization assay |
| Antibody | Anti-GFP Alexa Fluor 647 | BioLegend | FM264G | Surface staining for flow cytometry: 1:20 |
| Antibody | Anti-rabbit IgG Alexa Fluor 647 | Thermo Fisher | A27040 | Anti-IgG surface FACS control: 1:200 |
| Antibody | Mouse monoclonal anti-FLAG | Sigma-Aldrich | F1804 | IF (1:1000), WB (1:1000) |
| Antibody | Rabbit polyclonal anti-LAMP1 | Abcam | ab24170 | IF (1:250) |
| Antibody | Mouse monoclonal anti-Sec31A | BD | 612350 | IF (1:500) |
| Antibody | Mouse monoclonal anti-FLAG beads | Sigma-Aldrich | M8823 | 40 µL per immunoprecipitation |
| Recombinant DNA reagent (plasmid) | Xlone | Addgene | 96930 | Gift from Xiaojun Lian |

*Appendix 1 Continued on next page*

*Appendix 1 Continued*

| Reagent type (species) or resource | Designation | Source or reference | Identifiers | Additional information |
|---|---|---|---|---|
| Recombinant DNA reagent (plasmid) | DICE landing pad HDR template | This paper | pJM313 | AAVS1 attP-PGK-TK-P2A-TagBFP-attP |
| Recombinant DNA reagent (plasmid) | Dual integrase expression vector | Other | pEN476 | CAGGS-BxbI-P2A-PhiC31 gift from Dr. Elphege Nora |
| Recombinant DNA reagent (plasmid) | PiggyBac Transposase | System Biosciences | PB200A_1 | |
| Sequence-based reagent | 5' EMD guide RNA | Synthego | 11400580 | ACAGAUUGGCUAGCGGCAGG |
| Sequence-based reagent | 3' EMD guide RNA | Synthego | 11400580 | ACAUGGGAGAAAAGCUCCAA |
| Sequence-based reagent | AAVS1 guide RNA | Synthego | 11400580 | accaauccugucccuag |
| Peptide, recombinant protein | Benzonase | EMD-Millipore | 70664-10KUN | 1:1000 in lysis buffer |
| Peptide, recombinant protein | pyogenes WT Cas9-NLS, purified | Berkeley QB3 MacroLab | | 2 µg/12.5 pmol per 100 µL reaction |
| Peptide, recombinant protein | 3x FLAG peptide | MedChem Express | HY-P0319A | FLAG elution during immunoprecipitation |
| Commercial assay or kit | PreOmics iST kit | PreOmics | P.O.00027 | MS sample prep |
| Commercial assay or kit | NEB Site-Directed Mutagenesis Kit | New England Biolabs | E0554 | Site-directed mutagenesis |
| Commercial assay or kit | NEB HiFi DNA Assembly Kit | New England Biolabs | E5520 | High-efficiency DNA assembly |
| Commercial assay or kit | MMLV reverse transcriptase | New England Biolabs | M0253 | cDNA generation |
| Commercial assay or kit | ZymoPURE II Midiprep Kit | Zymo Research | D4202 | High-purity plasmid preparation |
| Chemical compound, drug | Blasticidin | Research Products International | B12150-0.1 | 5 µg/mL for U2OS selection |
| Chemical compound, drug | G418 (Geneticin) | Fisher Scientific | 501363278 | 400 µg/mL (U2OS selection), 200 µg/mL (U2OS maintenance) |
| Chemical compound, drug | Doxycycline | Sigma-Aldrich | D9891 | Inducible expression; 1–2 µg/mL |
| Chemical compound, drug | Bafilomycin A1 | Sigma-Aldrich | B1793 | Lysosome blocking agent; 100 nM |
| Chemical compound, drug | Puromycin | Millipore-Sigma | 540411-25mg | 0.25 µg/mL for hiPSC selection |
| Software, algorithm | Fiji | https://imagej.net | | Image and WB analysis |
| Software, algorithm | FlowJo | https://www.flowjo.com/ | | Flow cytometry analysis |
| Software, algorithm | Nikon Elements | https://www.microscope.healthcare.nikon.com/products/software/nis-elements | | Image acquisition |
| Software, algorithm | GraphPad Prism | https://www.graphpad.com | | Statistical tests |
| Software, algorithm | Jupyter Lab | https://jupyter.org/ | | Analysis and plotting of MS data |
| Other | McCoy's 5A Medium | Thermo Fisher | 16600082 | Culture medium for U2OS cells |
| Other | mTeSR+Medium | StemCell Technologies | 85850 | Culture medium for WTC-11 hiPSCs |
| Other | ReLeSR | StemCell Technologies | 100-0483 | Clump passaging of hiPSCs |
| Other | ROCK inhibitor | Selleck Chemicals | 101763-964 | 10 µM |
| Other | Accutase | Fisher Scientific | NC9464543 | Single-cell passaging of hiPSCs |

*Appendix 1 Continued*

| Reagent type (species) or resource | Designation | Source or reference | Identifiers | Additional information |
|---|---|---|---|---|
| Other | Matrigel | Corning | 354277 | Coating for hiPSC culture vessels |
| Other | Hoechst 33342 | Tocris Bioscience | 5117 | Used at 10 µg/mL |
| Other | Protease inhibitor cocktail | Sigma-Aldrich | P8340 | |
| Other | PhosSTOP phosphatase inhibitor | Roche | 4906845001 | |
| Other | Lipofectamine 2000 | Invitrogen | 11668019 | |

