## [Editor Report · eLife Assessment]

This study presents a **valuable** finding on the delivery of a nuclear envelop protein to lysosomes and the impact of C-terminal tagging on its traffic. The authors provide **solid** evidence for the potential artifacts introduced by large terminal tags, particularly in the context of membrane protein localization and stability.

---

## [Referee Report · Reviewer #1 (Public review)]

Summary:

The authors revisit the specific domains/signals required for redirection of an inner nuclear membrane protein, emerin, to the secretory pathway. They find that epitope tagging influences protein fate, serving as a cautionary tale for how different visualisation methods are used. Multiple tags and lines of evidence are used, providing solid evidence for the altered fate of different constructs.

Strengths:

This is a thorough dissection of domains and properties that confer INM retention vs secretion to the PM/lysosome, and will serve the community well as a caution regarding placement of tags and how this influences protein fate.

Weaknesses:

The specific biogenesis pathway for C-terminally tagged emerin might confound some interpretations. Appending the large GFP to the C-terminus may direct the fusion protein to a different ER insertion pathway than that used by the endogenous protein. How this might influence the fate of the tagged protein remains to be determined. In some ways this is beyond the scope of the current study, but should serve as a warning to epitope-tagging approaches.

---

## [Referee Report · Reviewer #2 (Public review)]

In this manuscript, Mella et al. investigate the effect of GFP tagging on the localization and stability of the nuclear-localized tail-anchored (TA) protein Emerin. A previous study from this group demonstrated that C-terminally GFP-tagged Emerin traffics to the plasma membrane and is eventually targeted to lysosomes for degradation. It has been suggested that the C-terminal tagging of TA proteins may shift their insertion from the post-translational TRC/GET pathway to the co-translational SRP-mediated pathway. Consistent with this, the authors confirm that C-terminal GFP tagging causes Emerin to mislocalize to the plasma membrane and subsequently to lysosomes.

In this study, they investigate the mechanism underlying this misrouting. By manipulating the cytosolic domain and the hydrophobicity of the transmembrane domain (TMD), the authors show that an ER retention sequence and increased TMD hydrophobicity contribute to Emerin's trafficking through the secretory pathway.

This reviewer had previously raised the concern that the potential role of the GFP tag within the ER lumen in promoting secretory trafficking was not addressed. In the revised manuscript, the authors respond to this concern by examining the co-localization of Emerin-GFP with the ER exit site marker Sec31A. Their data show that the presence of the C-terminal GFP tag increases Emerin's propensity to engage ER exit sites, supporting the conclusion that GFP tagging promotes its entry into the secretory pathway.

---

## [Author Response]

The following is the authors’ response to the original reviews.

**Reviewer #1 (Public review):**
Summary:The authors revisit the specific domains/signals required for the redirection of an inner nuclear membrane protein, emerin, to the secretory pathway. They find that epitope tagging influences protein fate, serving as a cautionary tale for how different visualisation methods are used. Multiple tags and lines of evidence are used, providing solid evidence for the altered fate of different constructs.Strengths:This is a thorough dissection of domains and properties that confer INM retention vs secretion to the PM/lysosome, and will serve the community well as a caution regarding the placement of tags and how this influences protein fate.Weaknesses:Biogenesis pathways are not explored experimentally: it would be interesting to know if the lysosomal pool arrives there via the secretory pathway (eg by engineering a glycosylation site into the lumenal domain) or by autophagy, where failed insertion products may accumulate in the cytoplasm and be degraded directly from cytoplasmic inclusions.

This manuscript is a Research Advance that follows previous work that we published in eLife on this topic (Buchwalter et al., *eLife* 2019; PMID 31599721). In that prior publication, we showed that emerin-GFP arrives at the lysosome by secretion and exposure at the PM, followed by internalization. While we state these previous findings in this manuscript, we did not explicitly restate here how we came to that conclusion. In the 2019 study, we (i) engineered in a glycosylation site, which demonstrated that emerin-GFP receives complex, Endo H-resistant N-glycans, indicating passage through the Golgi; (ii) performed cell surface labeling, which confirmed that emerin accesses the PM; and interfered with (iii) the early secretory pathway using brefeldin A and with (iv) lysosomal function using bafilomycin A1. Further, we ruled out autophagy as a major contributor to emerin trafficking by treating cells with the PI3K inhibitor KU55933, which had no effect on emerin’s lysosomal delivery.

It would be helpful if the topology of constructs could be directly demonstrated by pulse-labelling and protease protection. It's possible that there are mixed pools of both topologies that might complicate interpretation.

We demonstrate that emerin’s TMD inserts in a tail-anchored orientation (C terminus in ER lumen) by appending a GFP tag to either the N or C terminus, followed by anti-GFP antibody labeling of unpermeabilized cells (Fig. 1G). This shows the preferred topology of emerin’s wild type TMD.

As the reviewer points out, it is possible that our manipulations of the TMD sequence (Fig. 2D-E) alter its preferred topology of membrane insertion. We addressed this question by performing anti-GFP and anti-emerin antibody labeling of the less hydrophobic TMD mutant (EMD-TMDm-GFP) after selective permeabilization of the plasma membrane (Figure 2 supplement, panel F). If emerin biogenesis is normal, the GFP tag should face the ER lumen while the emerin antibody epitope should be cytosolic. If the fidelity of emerin’s membrane insertion is impaired, the GFP tag could be exposed to the cytosol (flipped orientation), which would be detected by anti-GFP labeling upon plasma membrane permeabilization. We find that the C-terminal GFP tag is completely inaccessible to antibody when the PM is selectively permeabilized with digitonin, but is readily detected when all intracellular membranes are permeabilized with Triton-X-100. These data confirm that mutating emerin’s TMD does not disrupt the protein’s membrane topology.

**Reviewer #2 (Public review):**
In this manuscript, Mella et al. investigate the effect of GFP tagging on the localization and stability of the nuclear-localized tail-anchored (TA) protein Emerin. A previous study from this group showed that C-terminally GFP-tagged Emerin protein traffics to the plasma membrane and reaches lysosomes for degradation. It is suggested that the C-terminal tagging of tail-anchored proteins shifts their insertion from the post-translational TRC/GET pathway to the co-translational SRP-mediated pathway. The authors of this paper found that C-terminal GFP tagging causes Emerin to localize to the plasma membrane and eventually reach lysosomes. They investigated the mechanism by which Emerin-GFP moves to the secretory pathway. By manipulating the cytosolic domain and the hydrophobicity of the transmembrane domain (TMD), the authors identify that an ER retention sequence and strong TMD hydrophobicity contribute to Emerin trafficking to the secretory pathway. Overall, the data are solid, and the knowledge will be useful to the field. However, the authors do not fully answer the question of why C-terminally GFP-tagged Emerin moves to the secretory pathway. Importantly, the authors did not consider the possible roles of GFP in the ER lumen influencing Emerin trafficking to the secretory pathway.
**Reviewer #2 (Recommendations for the authors):**
Major concerns:(1) The authors suggest that an ER retention sequence and high hydrophobicity of Emerin TMD contribute to its trafficking to the secretory pathway. However, these two features are also present in WT Emerin, which correctly localizes to the inner nuclear membrane. Additionally, the authors show that the ER retention sequence is normally obscured by the LEM domain. The key difference between WT Emerin and Emerin-GFP is the presence of GFP in the ER lumen. The authors missed investigating the role of GFP in the ER lumen in influencing Emerin trafficking to the secretory pathway. It is likely that COPII carrier vesicles capture GFP protein in the lumen as part of the bulk flow mechanism for transport to the Golgi compartment. The authors could easily test this by appending a KDEL sequence to the C-terminus of GFP; this should now redirect the protein to the nucleus.

We agree with the reviewer’s point that the presence of lumenal GFP somehow promotes secretion of emerin from the ER, likely at the stage of enhancing its packaging into COPII vesicles. We struggle to think about how to interpret the KDEL tagging experiment that the reviewer proposes, as the KDEL receptor predominantly recycles soluble proteins from the Golgi to the ER, while emerin is a membrane protein; and we have shown that emerin already contains a putative COPI-interacting RRR recycling motif in its cytosolic domain.

Nevertheless, we agree with the reviewer that it is worthwhile to test the possibility that addition of GFP to emerin’s C-terminus promotes capture by COPII vesicles. We have evaluated this question by performing temperature block experiments to cause cargo accumulation within stalled COPII-coated ER exit sites, then comparing the propensity of various untagged and tagged emerin variants to enrich in ER exit sites as judged by colocalization with the COPII subunit Sec31a. These data now appear in Figure 4 supplement 1. These experiments indicate that emerin-GFP samples ER exit sites significantly more than does untagged emerin. Further, the ER exit site enrichment of emerin-GFP is dampened by shortening emerin’s TMD. We do not see further enrichment of any emerin variant in ER exit sites when COPII vesicle budding is stalled by low temperature incubation, implying that emerin lacks any positive sorting signals that direct its selective enrichment in COPII vesicles. Altogether, these data indicate that both emerin’s long and hydrophobic TMD and the addition of a lumenal GFP tag increase emerin’s propensity to sample ER exit sites and undergo non-selective, “bulk flow” ER export.

(2) The authors nicely demonstrate that the hydrophobicity of Emerin TMD plays a role in its secretory trafficking. I wonder if this feature may be beneficial for cells to degrade newly synthesized Emerin via the lysosomal pathway during mitosis, as the nuclear envelope breakdown may prevent the correct localization of newly synthesized Emerin. The authors could test Emerin localization during mitosis. Such findings could add to the physiological significance of their findings. At the minimum, they should discuss this possibility.

We thank the reviewer for this insightful suggestion. It is attractive to speculate that secretory trafficking might enable lysosomal degradation of emerin during mitosis, when its lamin anchor has been depolymerized. However, we think it is unlikely that mitotic trafficking contributes significantly to the turnover flux of untagged emerin; if it did, we would expect to see higher steady state levels and/or slowed turnover of emerin mutants that cannot traffic to the lysosome. We did not observe this outcome. Instead, mutations that enhance (RA) or impair (TMDm) emerin trafficking had no effect on the untagged protein’s steady-state levels (Fig. 4G).

Minor concerns:(1) On page 7, the authors note that "FLAG-RA construct was not poorly expressed relative to WR, in contrast with RA-GFP (Figures S3C, 2I)." The expression levels of these proteins cannot be compared across two different blots.

We apologize for this confusion; we were implying two distinct comparisons to internal controls present on each blot. We have adjusted the text to read “FLAG-RA construct was not poorly expressed relative to FLAG-WT (Fig. S3C) in contrast to RA-GFP compared to WT-GFP (Fig. 2I).”

(2) In the first paragraph of the discussion, the authors suggest that aromatic amino acids facilitate trafficking to lysosomes. However, they only replaced aromatic amino acids with alanine residues. If they want to make this claim, they should test other amino acids, particularly hydrophobic amino acids such as leucine.

The reviewer may be inferring more import from our statement than we intended. We focused on these aromatic residues within the TMD because they contribute strongly to its overall hydrophobicity. Experimentally, we determined that nonconservative alanine substitutions of these aromatic residues inhibited trafficking. We do not state and do not intend to imply that the aromatic character of these residues specifically influences trafficking propensity, and we agree with the reviewer that to test such a question would require additional substitutions with non-aromatic hydrophobic amino acids.

We realize that our phrasing may have been misleading by opening with discussion of the aromatic amino acids; in the revised discussion paragraph, we instead lead with discussion of TMD hydrophobicity, and then state how the specific substitutions we made affect trafficking.

Reviewing Editor comments:While reviewer 1 did not provide any recommendations to the authors, I agree with this reviewer that the authors should validate the topology of their tagged proteins (at least for the one used to draw key conclusions). Given that Emerin is a tail-anchored protein, having a big GFP tag at the C-terminus could mess up ER insertion, causing the protein to take a wrong topology or even be mislocalized in the cytosol, particularly under overexpression conditions. In either case, it can be subject to quality control-dependent clearance via either autophagy, ERphagy, or ER-to-lysosome trafficking. I think that the authors should try a few straightforward experiments such as brefeldin A treatment or dominant negative Sar1 expression to test whether blocking conventional ER-to-Golgi trafficking affects lysosomal delivery of Emerin. I also think that the authors should discuss their findings in the context of the RESET pathway reported previously (PMID: 25083867). The ER stress-dependent trafficking of tagged Emerin to the PM and lysosomes appears to follow a similar trafficking pattern as RESET, although the authors did not demonstrate that Emerin traffic to lysosomes via the PM. In this regard, they should tone down their conclusion and discuss their findings in the context of the RESET pathway, which could serve as a model for their substrate.

We agree that validating the topology of TMD mutants is important, and now include these experiments in the revised manuscript (please see our response to Reviewer 1 above).

Please see our response to Reviewer 1’s public review; we previously determined that emerin-GFP undergoes ER-to-Golgi trafficking (see our 2019 study).

We recognize the major parallels between our findings and the RESET pathway. In our 2019 study, we found that similarly to other RESET cargoes, emerin-GFP travels through the secretory pathway, is exposed at the PM, and is then internalized and delivered to lysosomes. We discussed these strong parallels to RESET in our 2019 study. In this revised manuscript, we now also point out the parallels between emerin trafficking and RESET and cite the 2014 study by Satpute-Krishnan and colleagues (PMID 25083867)